# A Histological and Morphometric Assessment of the Adult and Juvenile Rat Livers after Mild Traumatic Brain Injury

**DOI:** 10.3390/cells10051121

**Published:** 2021-05-06

**Authors:** Ruslan Prus, Olena Appelhans, Maksim Logash, Petro Pokotylo, Grzegorz Józef Nowicki, Barbara Ślusarska

**Affiliations:** 1Department of Normal and Pathological Clinical Anatomy, Odessa National Medical University, UA-65000 Odessa, Ukraine; ruslan.prus@onmedu.edu.ua (R.P.); olena.appelhans@onmedu.edu.ua (O.A.); 2Department of Normal Anatomy, Lviv National Medical University, UA-79010 Lviv, Ukraine; anatompetro@meduniv.lviv.ua; 3Department of Family Medicine and Community Nursing, Medical University of Lublin, PL-20-081 Lublin, Poland; gnowicki84@gmail.com (G.J.N.); barbara.slusarska@umlub.pl (B.Ś.)

**Keywords:** traumatic brain injury, mild traumatic brain injury, liver, juvenile rats, extracranial complication

## Abstract

Traumatic brain injury (TBI) is one of the most severe problems of modern medicine that plays a dominant role in morbidity and mortality in economically developed countries. Our experimental study aimed to evaluate the histological and morphological changes occurring in the liver of adult and juvenile mildly traumatized rats (mTBI) in a time-dependent model. The experiment was performed on 70 adult white rats at three months of age and 70 juvenile rats aged 20 days. The mTBI was modelled by the Impact-Acceleration Model-free fall of weight in the parieto-occipital area. For histopathological comparison, the samples were taken on the 1st, 3rd, 5th, 7th, 14th, and 21st days after TBI. In adult rats, dominated changes in the microcirculatory bed in the form of blood stasis in sinusoidal capillaries and veins, RBC sludge, and adherence to the vessel wall with the subsequent appearance of perivascular and focal leukocytic infiltrates. In juvenile rats, changes in the parenchyma in the form of hepatocyte dystrophy prevailed. In both groups, the highest manifestation of the changes was observed on 5–7 days of the study. On 14–21 days, compensatory phenomena prevailed in both groups. Mild TBI causes changes in the liver of both adult and juvenile rats. The morphological pattern and dynamics of liver changes, due to mild TBI, are different in adult and juvenile rats.

## 1. Introduction

Traumatic brain injury (TBI) in children and young people is one of the most acute problems of modern medicine that can lead to death, long-term complications, or life-long disability [1,2]. Traumatic brain injury affects over 10 million people in the world every year [3], including 2.5 million people in Europe [4]. Since 2007, the number of children presented to Hospital Emergency Departments, due to traumatic brain injury, has increased. Reports from the United States (USA) indicate that approximately 1591.5 out of 100,000 children aged 0–4 years suffer from a traumatic brain injury. The percentage of children aged 15–24 with brain injuries is 1080.7 per 100,000 children [5]. Paediatric TBI covers a wide range of age groups with different injury mechanisms and different mortality rates [6]. Epidemiological studies focusing on the mechanisms of brain injuries in children have shown that violence-related brain injuries are widespread in children under two years of age [7]; that they are caused by a fall in children aged 5–14, whereas traffic accidents most often caused brain injury in children over 15 years of age [5,8].

Mild Traumatic Brain Injury (mTBI), commonly known as a concussion, is the most common traumatic brain injury [9]. The Mild Traumatic Brain Injury Committee of the American Congress of Rehabilitation Medicine [10] defines mTBI as a mild head injury that causes a short period of unconsciousness followed by cognitive impairment. Along with cognitive impairment, mTBI causes various other symptoms, including headache, fatigue, depression, anxiety, and irritability, collectively known as post-concussion syndrome (PSC). mTBI usually affects the frontal and temporal lobes of the brain, which are related to executive function, learning, and memory [11]. As these areas continue to develop in childhood, children are particularly vulnerable to the adverse effects of mTBI [12].

TBI causes tissue damage within the central nervous system (CNS), followed by an intense and rapid inflammatory response that may lead to permanent inflammation [13,14]. This inflammatory response is not limited to the brain. Inflammatory mediators are released from damaged brain tissue into the bloodstream, causing a systemic inflammatory response in peripheral organs. The identity of many of these mediators is still unclear, although TBI is known to have a significant effect on several peripheral organs, including the lungs, spleen, and liver [15,16,17,18]. TBI induces an acute-phase response (APR) mainly under hepatic control [19]. APR initiated primarily by circulating cytokines (IL-1β, IL-6, and TNFα) [18] can lead to a coordinated liver transcription response program [20] involving the potent activation of NF-κB regulated acute phase proteins and the release of multiple cytokines and chemokines into the bloodstream [21]. Anthony et al. [15] showed that hepatic chemokine production acts as an enhancer of the focal injury response, providing the CNS-liver communication pathway. However, how these inflammatory mediators contribute to the deleterious effects of TBI remains unknown [22]. One way to understand the influence of pro-inflammatory mediators on the harmful effects of TBI is to understand the histological and morphometric changes that occur in liver cells, as a result of TBI. Unfortunately, no experimental studies would confirm or disprove the assumptions concerning morphological changes in the liver due to TBI, especially mTBI. The difference between these changes and age also remains undiscovered. Therefore, our experimental research aimed to evaluate the histological and morphological changes occurring in the liver of adult and juvenile mildly traumatized rats (mTBI) in a time-dependent model.

## 2. Materials and Methods

### 2.1. Animals and Experimental Conditions

Wister Albino rats (*n* = 140) were used as model organisms in this study. The rats were divided into two equal groups: 70 sexually mature white males aged three months (weight = 180–230 g) and 70 young males aged 20 days (weight = 20–25 g). The rats were bred in a vivarium at the National Medical University of Odessa. All the animals used in the experiment were kept in the vivarium of the Odessa National Medical University. All rats were housed separately in propylene cages (dimensions: 33 cm × 19 cm × 14 cm) at an optimal temperature of 22–24 °C on a standard light-dark cycle (LD; 12:12 light (~150 lux)/dark (0 lux) for 4 weeks of the experiment. The animals had unlimited access to feed (PK 120-1 laboratory animal feed mixture produced by “Rezonivski korma”) and filtered tap water ad libitum. All experimental procedures were approved by the Bioethics Committee of the National Medical University of Odessa (Protocol No. 109-A of 4 November 2019) and were carried out in accordance with the provisions of the European Convention for the Protection of Vertebrates Used for Experimental and Other Purposes (Strasbourg, 18 March 1986), the resolution of the First National Congress of Ukraine on bioethics (2001) and the Order of the Minister of Public Health of Ukraine No. 690 of 23/09/2009. The study was conducted following the guidelines of the Helsinki Declaration and in accordance with the regulations on the care of animals. After one week of acclimatization, the rats were randomly divided into four subgroups: Group I: adult rats (*n* = 70), subgroup IA: control group (*n* = 10), subgroup IB: animals with model mild mechanical traumatic brain injury (*n* = 60). Group II: young rats (*n* = 70), subgroup IIA: control group (*n* = 10), subgroup IIB: animals with model mild mechanical brain trauma (*n* = 60).

### 2.2. Induction of a Mild Traumatic Brain Injury

Mild traumatic brain injury was caused by the Impact Acceleration Model-free weight loss in the parieto-occipital area of the rat, according to the patent by Meretskyi et al. [23]. For mTBI induction, a stand with a vertical metal tube, 1 mm in diameter and 65 cm high, was attached. A metal tube served as a guide for the free-falling weight, a round metal bar, to the lower end of which (the place of impact) a rubber strip 3 mm thick with an area of 0.5 cm^2^ was glued. The device stand was attached to the ground. The rats were lightly anesthetized with ether (until they were unresponsive to paw or tail pinches) and then placed under the weight. The weight fell freely, hitting the rat in the parieto-occipital area. The center of impact was on the shooting line 3–5 mm below the intradermal line. To induce mTBI in adult male rats, a weight of 34.5 g was used, which generated a shock energy of 0.22 J. In young rats, a weight of 15 g was used, which produced a shock energy of 0.112 J [24] (Appendix A).

### 2.3. Experiment Design

The experiment began after a week of the rats’ acclimatization. The rats were divided into the following groups:Group I (*n* = 70; sexually mature males):
-Group IA (*n* = 10; control group),-Group IB (*n* = 60; animals with model mild traumatic brain injury)
Group II (*n* = 70; young males):
-Group IIA (*n* = 10; control group),-Group IIB (*n* = 60; animals with a model mild traumatic brain injury)

The details of the experimental design are presented in Figure 1. In the studied groups of rats, mTBI was induced to varying degrees depending on age. Animals from groups IB and IIB were removed randomly from the study on days 1, 3, 5, 7, 14, and 21 after induced mTBI (10 animals each). On the other hand, animals from the control groups (IA and IIA) were divided into two subgroups (5 pieces each) and removed from the experiment after one week of acclimatization and at the end of the experiment (in the fourth week of the experiment). The results of the observations obtained from groups IB and IIB with IA and IIA were compared. Rats were removed from the experiment by overdosing on ether anesthesia.

### 2.4. Histomorphometric Analysis

Samples for the histology were taken from each lobe of the liver by its longitudinal, vertically orientated section at a random angle. The second cut was made parallel to the first at a 3–5 mm distance to obtain a thin longitudinal sample. Taken samples of a 3–5 mm thickness were fixed in 10% neutral buffered formalin for 24 h. After that, samples were dehydrated through an ethanol series (from 60% to 100%), cleared with ethanol-xylene (50%–50%) and xylene (100%), and embedded into paraffine vax. Serial 5 µm paraffin sections were made with a rotating microtome. Each sample was made with 1 glass, stained by Hematoxylin and Eosin, and 2 samples from each animal were stained by Mallory’s trichrome and VanGieson’s, so 6 glasses obtained the description from each animal. For the morphometric study, 2 glasses out of 4 with H&E (from each animal) were randomly selected. The histological glass slides were analysed and photo-documented with the microscopes “Leica-DMLS” and “Meiji MT4300 LED” with a Canon EOS 550D camera (using MA150/50 and MA986 adapters).

Additionally, “Darktable” software was used. A series of images were taken in a uniform step from one edge of the histology sample to another, starting at a random point (systematic uniform random sampling). All measurements were performed on images taken at a magnification of ×400. All morphometric studies were performed exclusively with primary, unedited Jpeg photographs with a resolution of 5184 × 3456 with the same calibration data for each magnification (pixel to μm ratio). The calibration for the morphometry was performed using a slide Meiji MA285 to determine the ratio of pixel to micrometer. All linear measurements and calibrations were performed with ImageJ ver.1.51j8 [25,26,27] using a straight-line tool at the appropriate calibration. To measure the diameter of the sinusoidal capillaries by the method of orthogonal intercepts (without the correction factor) [28,29], a grid of 1000 μm cell lines was applied to the image using a “Grid” tool with random offset to determine the point of intersection of the lines with the object as the unbiased primary point of the measurement (Appendix A). A perpendicular line was drawn from it to the opposite side of the object. After the pilot measurements, the optimal sample size for each period (at least 200) and the minimum number of analyzed areas for each period (at least 10) were established. Cytomorphometry (hepatocytes diameter) was performed by a simplified version of the “nucleator” method [30]. In this method, as an unbiased point for measurement, a nucleolus was used with a line extension to the cell boundary, the number of measurements from each cell—4, the direction of the first measurement was determined randomly, the next three—at an angle of 90° to every previous one.

### 2.5. Statistical Analyses

The obtained results were tested for the normality of distribution. All data with a normal distribution were presented as M ± SD, where M indicates the mean average and SD stands for the standard deviation. The difference between the groups was analyzed using a t-test (for two groups—comparison between Group I and Group II for each term) and an analysis of variance (ANOVA) with Tukey’s posthoc test (for more than two groups—comparison of Group I or Group II subgroups during the experiment). In cases where data were not normally distributed, the median and quartile were used to describe the sample, and nonparametric tests (Kruskal–Wallis H-test with posthock Dunn’s test, Mann–Whitney U test) were used for comparison. The differences were considered statistically significant with *p* < 0.05. All statistical computing was made with RStudio ver. 1.1.442 software.

## 3. Results

### 3.1. Histological Examinations of Rat Livers on the First Day after Injury

Histological examination of the livers of adult rats one day after the mTBI revealed a pronounced reaction of the vascular bed, manifested as the blood stasis of the branches of the portal vein, sinus capillaries, and more rarely, branches of the hepatic artery and central veins. These were accompanied by red blood cells (RBC) sludges and the aggregation of RBC into the vessels’ walls and diapedesis. In the morphometric study, the diameter of the sinusoidal capillaries was increased to 6.03 μm (Table 1 and Appendix A). In some areas, the dilatation of capillaries greater than 15 microns with a maximum value of 40.06 μm was noted. The most dilated capillaries were observed in the subcapsular area (Figure 2). The increase in the diameter of sinusoidal capillaries on day 1 of the experiment was statistically significant compared to the control group (*p* < 0.001). Histological examination of the liver of juvenile rats 1 day after the mTBI revealed a reaction of the vascular bed, similar to that in adult rats. Such dynamics are confirmed by the tendency for the diameters of sinusoidal capillaries to increase their value was 3.72 μm, although the increase was not statistically significant (*p* = 0.61).

### 3.2. Histological Examinations of Rat Livers on the Third Day after Injury

On the 3rd day after a mild traumatic brain injury, there was a tendency for vascular disorders in the liver of adult rats to have increased. Besides, there were focal changes of the trabecular structure up to fully discomplexed hepatocytes were observed, as well as rarely focal dystrophy of hepatocytes with the occurrence of pycnotic nuclei. Typical for this term was the appearance around some portal triads of lymphocytic and Kupfer-cells infiltrates, which in some cases reached a diameter of 40–60 μm. The blood stasis of the sinusoidal capillaries, which in the previous term was mainly expressed in areas close to the liver capsule, was now observed in the depth of the parenchyma (Figure 3). The morphometry of sinusoidal capillaries confirms this tendency for changes of the microcirculatory bed, so on the 3rd day of the study, their diameter was 9.40 μm, which was significantly higher in both the control group (*p* < 0.001) and in the previous term of the experiment (*p* < 0.001).

Three days after mild TBI in the liver histology of juvenile rats, a large number of hepatocytes with dystrophy were revealed. The cytoplasm of such cells is enlightened, irregularly stained, or contains no dye at all with medium-sized vacuoles. The localization of such cells is mainly a subcapsular zone and diffusely deep into the parenchyma in the form of small clusters of 50–100 cells (Figure 4). Changes in the microcirculatory bed in this term appear similar to the previous one—blood stasis in the portal veins and sinusoidal capillaries with RBC sludges and their aggregation to the vessel walls. However, the sinusoidal capillaries’ diameter decreased compared to the previous term and with the control group value and reached 2.38 μm.

### 3.3. Histological Examinations of Rat Livers on the Fifth Day after Injury

Five days after mTBI, a histologic examination of changes in the vascular bed of the liver revealed no pronounced dynamics, compared with the previous term. Thus, the diameter of the sinusoidal capillaries was 10.17 μm, which, despite a further slight upward trend, had no significant difference in comparison to the previous term (*p* = 0.53) but was significantly higher than the control group (*p* < 0.001). Typical for this term is a greater number of lymphocytic perivascular infiltrates around large vessels and, in particular, periportal infiltrates, whose thickness ranges from 20 μm up to 70 μm. As in the previous term, a large number of dilated capillaries with a diameter of more than 20 μm were observed with blood stasis, RBC sludges, and thrombus formation near the vessel wall. Another characteristic of this term is the change in the structure of the lobules, particularly the presence of large sections of the parenchyma with a chaotic arrangement of hepatocytes without the classic liver trabecular structure.

On the 5th day after mild TBI in the juvenile rats’ liver, histological samples continued to reveal the previous terms’ increased changes. Thus, the overwhelming number of hepatocytes on examined samples were with signs of dystrophy in the form of an enlightened or emptied cytoplasm. A large number of cells with pycnotic nuclei were observed. Focal necrosis occurs. The trabecular structure of the liver in some areas was disturbed; most hepatocytes in such areas are smaller in size and arranged erratically. The blood stasis was observed in the portal and central veins (Figure 5). The sinusoidal capillaries were straight in shape; their diameter had slightly increased, compared to the previous term (*p* = 0.01) and amounted to 2.64 μm; however, they remained significantly low in comparison with the control group (*p* < 0.001). Some sinusoidal capillaries dilated up to 10 microns; in their lumens, RBC sludges and blood stasis were revealed.

### 3.4. Histological Examinations of Rats’ Liver on the Seventh Day after Trauma

A histological examination of the liver of rats on day 7 after mild TBI revealed a tendency for a decrease in the intensity of vascular disorders. Moreover, there are signs of compensation for the disturbances that have been identified in the previous terms.

At that time, vascular disorders continued to become diffuse. Thus, there was an increase in plasma impregnation with fibrous swelling in the walls of small vessels. The blood stasis of hepatic triads of all calibers and sinusoidal capillaries with RBC sludges and clots, some with complete occlusion of the lumen, was retained. There were still signs of hepatocyte dystrophy in the peripheral areas of the liver lobules, which were accompanied by focal necrosis. However, as in the previous term, this phenomenon had not acquired a systemic character. Infiltrates are also observed at the sites of previous focal necrosis; fibroblasts and collagen fibers were found in such areas (Figure 6). The diameter of the sinusoidal capillaries slightly decreased at 7 days of the experiment, amounting to 8.37 μm (was significantly lower than the previous term (*p* < 0.001), while remaining significantly higher than the control group (*p* < 0.001)). Dilated sinusoidal capillaries were also less common—a maximum of 29.18 μm was observed.

Seven days after mild TBI, the histology examination of juvenile rats’ liver revealed uneven changes in different samples. So, in some samples, a large number of hepatocytes with light cytoplasm and vacuoles, which sometimes form massive clusters, were found. Sometimes such clusters are localized under the capsule of the liver, occasionally in-depth in the form of ring-shape foci at the periphery of the hepatic lobules or in the 3rd zone of the classic hepatic lobule, sometimes all zones of the lobule contain dystrophic hepatocytes (Figure 7). The diameter of such cells ranges from 25–33 μm; the maximum value was 35.24 μm. Clusters of small young hepatocytes border such foci with a diameter of 10–15 μm. The heterogeneity of changes in cell diameters leads to the average value remaining close to that of the control group and is 17.44 ± 6.12 μm (16.15 ± 4.16 μm in the control group, *p* = 0.065). Single nodal necrosis with diapedesis occurs. In areas with dystrophic hepatocytes, the sinusoidal capillaries are narrowed to 1.4 μm, or complete occlusion is observed. Some sinusoidal capillaries are dilated, with a maximum observed value of 15.45 μm. In general, as in the previous term, sinusoidal capillaries are straightened; their diameter also remains at the level of the previous term, 2.60 μm, which is significantly less than the corresponding indicator of the control group (*p* < 0.001). Changes are uneven and depend on the size of the lobes—smaller lobes show more pronounced hemodynamic changes, whereas larger parts of the liver manifest more dystrophic hepatocytes with vacuoles and less pronounced hemodynamic changes while, at the same time, deeper in the depth of larger lobes—unchanged hepatocytes with undisturbed liver structure occur.

### 3.5. Histological Examinations of the Livers of Rats on the Fourteenth and Twenty-First Days after Trauma

A pattern of compensatory changes predominated on the 14th and 21st days after mild TBI on the rat’s liver histology. In particular, the area of hepatocytes with dystrophic changes is reduced. Instead, a large number of areas with small, young hepatocytes with a diameter of 12–17 microns, with a uniformly colored, eosinophilic cytoplasm, appear. The manifestations of changes in the microcirculatory bed also decreased—plethora was recorded only sporadically in a few branches of the portal vein. In some veins, including the central veins, adhesion of erythrocytes to the vessel wall was observed. Sinusoidal capillaries are more tortuous in shape than in the previous period. Their diameter also increases—up to 3.16 µm on day 14 and 3.19 μm on day 21, close to that of the control group (*p* = 0.052). There is a thickening of the portal triad vessels’ walls with mild perivascular oedema and single lymphocytes in the periportal space, but these changes are not systemic (Figure 8).

Characteristic of these terms in adult animals, in addition to parietal thrombi in sinusoidal capillaries, lymphocytes, leukocytes, and macrophages in the lumen, was the appearance of collagen, detected by staining according to Mallory and Van Gieson (Figure 9). This phenomenon was most often observed around elements of portal triads and central veins. In some samples, collagen invasion was observed in the direction of the parenchyma, but the tendency to form a porto-central or porto-portal septum was not observed. At the same time, such a pattern was not observed in young rats (Figure 10 and Figure 11).

## 4. Discussion

The study evaluates the histological and morphometric changes in the liver in juvenile and adult rats after mTBI induction. Many researchers argue that TBI is a potent stress factor for the whole body, and therefore, due to a large number of cytokines, it triggers stress responses that start a chain of events. First, the sympathoadrenal system is activated. Second, this activation increases the concentration of catecholamines in the blood, cerebrospinal fluid, and urine. Catecholamines are directly involved in regulating cytokines, and elevated levels appear to influence the immune system during stress. Third, the increases lead to various vegetal-visceral reactions with manifestations of arterial hypertension (later-hypotension), increased basic metabolism, hyperthermia, increased protein catabolism, etc. [31,32,33,34,35]. After brain injury, the central dysregulation mechanisms could contribute to the development and progression of extracerebral organ dysfunction by promoting systemic inflammation that may cause medical complications. It has been shown that the dysregulation of catecholamines and glucocorticoids causes damage to cells with the involvement in the pathological process of the neuro-humoral apparatus, which in turn leads to an imbalance of the internal environment of the organism with the impaired activity of almost all organs and systems, increased endothelial permeability, and the expression of endothelial adhesion molecules [36].

In the liver of adult rats during our study, in the experiment, changes in the microcirculatory bed appeared, such as the blood stasis of the central and portal veins and sinusoidal capillaries, RBC sludges, and aggregation to the vessels wall. These changes were accompanied by massive perivascular and periportal lymphocytic and macrophage infiltration. These changes can be explained as a non-specific reaction caused by TBI, which produces a generalized effect on the organism—a complex of pathophysiological and pathomorphological changes, not only in the area of immediate mechanical injury but, in various organs and systems of the organism, particularly in the liver [37,38,39,40]. Changes in the microcirculatory bed of adult rats reached their maximum degree of manifestation on the 5th day following TBI, after which they had a stable tendency to decrease. In the stroma of the portal tracts, the collagen deposits were observed in the last periods, the walls of the portal tract vessels were thickened, due to collagen fibers. The activity of lymphocytes and macrophages and changes in endothelial cells, in the form of swelling of the nucleus and cytoplasm, were also characteristic of the liver of mature rats. However, dystrophic changes in the parenchyma cells of adult rats were somewhat sporadic and could not be considered tendentious. The activation of the cytokine system can initiate immune cell activity in response to brain damage in combination with microcirculation disturbance. This process can lead to oxidative stress in liver hepatocytes and their apoptosis.

In the liver of juvenile rats, changes in the hepatocytes prevailed, which was manifested by their dystrophy along with the presence of foci of active regeneration. Changes in the microcirculation were also observed in juvenile rats; however, only the plethora of veins and sinusoidal capillaries with blood stasis were common with adult rats. In contrast to that in adult animals, the diameter of sinusoidal capillaries of juvenile rats tended to decrease during the experiment, compared with the control group, except for day 1 of the study. This can be interpreted as the primary dilation of the sinusoidal capillaries of the liver on day 1, due to an increase in cardiac output as a result of theactivation of the stress mechanism of the cardiovascular system in response to trauma. Further reduction in the diameter of the sinusoidal capillaries was probably due to the appearance of large numbers of large dystrophic hepatocytes and focal disturbance of the cytoarchitecture of the liver, which could have caused their compression and impaired blood flow. Focal dilation and straightening capillaries in sites of preserved cytoarchitecture, indicating a local increase in blood pressure in sinusoidal capillaries, can be regarded as evidence in favor of this theory. Simultaneously, massive perivascular and focal lymphocytic infiltrates were not characteristic of changes in the liver of juvenile rats, unlike in adult animals. Such differences may indicate a fundamentally different response of the immune system to TBI in juvenile and adult rats, which may be explained by both the immaturity of the young rat’s immune system and the different activity of the cytokine system as the primary mechanism of systemic inflammatory processes, due to TBI. This feature of the microcirculation response of young animals may be related to the age characteristics of the sympathoadrenal and endocrine systems, which is characteristic of both animals and humans [41]. In particular, there is a fundamental functional difference between the limbic-hypothalamic-pituitary-adrenal axis in rats under two weeks of age [42], which affects the overall balance of gluco- and mineralocorticoids, the mechanisms of their regulation during stress and after. This period is characterized by a reduced response to stress with the subsequent inability to restore glucocorticoids’ balance quickly. The limbic-hypothalamic-pituitary-adrenal system is the main structure that regulates the stress-induced response of the body. From many aspects of its functioning, we are interested, first of all, in the activation of corticotropin-releasing factor and vasopressin secretion into the portal pituitary system. As a result, the secretion of the adrenocorticotropic hormone is activated, which stimulates the activity of the adrenal cortex, changing the balance in the direction of increasing the secretion of glucocorticoids (cortisol). The release of glucocorticoids helps to adjust metabolic processes in response to a stressful situation. These hormones are active immunosuppressants. Glucocorticoids, which are a powerful mechanism for regulating many functions in adults, are also involved in regulating growth and development in immature individuals [43], mainly responsible for the maturation and development of the brain. Therefore, a sharp increase in glucocorticoid levels in young animals can lead to catabolic changes, which against the background of immune suppression, can lead to increased posttraumatic endotoxication and greater manifestation of parenchymal changes compared to adult animals, as cortisol is a powerful immunosuppressant [44,45,46,47]. It should be borne in mind that early stress, as an isolated factor, can lead to further cardiac, metabolic, endocrine, and mental complications [48,49] even without considering the factor of trauma.

The nature of hepatocyte dystrophy in juvenile rats is difficult to explain, based only on routine staining with hematoxylin-eosin—other techniques must be used for this purpose. Numerous vacuoles in juvenile rat’s hepatocytes cytoplasm can be protein or lipid in content, but also numerous cells have hydropic dystrophy. Many factors mentioned before can cause such changes with liver parenchyma. Also, factors that affect the complications of TBI in children, such as the intensity of metabolic processes, low tolerance to blood loss, hypoxia and hypotension, and the prevalence of generalized reactions over local, high compensatory opportunities with a fast transition to decompensation, should be taken into account [50].

When analyzing the coverage in the modern literature of extracranial complications of TBI, most of the sources we have encountered describe changes in the organs in moderate and severe TBI [51,52,53]. Simultaneously, changes in cognitive functions and psycho-emotional state are mainly described in mild TBI [54,55,56]. Our results signalto search for a possible connection between the pathology of the internal organs and previous TBI in children and adolescents, even with a mild TBI. In this context, it should also be emphasized that the diagnosis of mTBI, made on the basis of classical signs, defined as a loss of consciousness for up to 30 min and a lower Glasgow Coma Scale score of 13–15, does not exclude intracranial changes, which raises the question of CT examination even in case of mTBI [57,58] and therefore does not exclude the possibility of extracranial morphological changes.

The strengths and weaknesses of this study require consideration. First of all, it is one of the few experimental studies describing changes in adult and juvenile rats’ liver cells after mTBI. Secondly, the changes taking place in liver cells presented by us may constitute the basis for an attempt to explain the CNS-liver communication pathway through mediators of the inflammatory response. Finally, third, the cellular changes in the liver described by us may lead to the production of inflammatory mediators and thus the regulation of the brain inflammatory response, and their inhibition may become a therapeutic target in patients with TBI and mTBI.

The limitation of full conclusions to be drawn from the results of this work lie in is its exclusively morphological character. For a full and comprehensive conclusion, the physiological aspects of these changes also need to be investigated to evaluate a possible strategy for treating and preventing liver complications from mTBI. Moreover, the fast development of young rats in combination with long-term experiments can be a limitation too. If the 21st day is the beginning of an independent life for rats after weaning from the mother, then 25–35 days is early adolescence, and 36–50 means it is an adolescent [41,59]. Another limitation of our research is the lack of marking, inter alia, pro-inflammatory cytokines in plasma and liver, prostaglandins or markers of oxidative stress in plasma or liver cells. Unfortunately, budget constraints prevented us from performing these determinations using the ELISA method, whereas performing the determinations using the immunohistochemical method was beyond the laboratory’s capabilities.

## 5. Conclusions

In conclusion, the present study revealed that mTBI influences morphological changes in the liver of young and adult rats. The dynamics of these changes are different in the two studied groups. The described changes in the liver cells can regulate the inflammatory response in the brain by producing cytokines and chemokines. The release of chemokines in the liver enhances the local response to injury by increasing the migration of circulating immune cells in the blood to the injured brain. However, the recruitment of leukocytes into the brain may also depend on the production of chemokines in the liver and APR, making their inhibition a valid therapeutic target in patients with TBI and mTBI.

## Figures and Tables

**Figure 1 cells-10-01121-f001:**
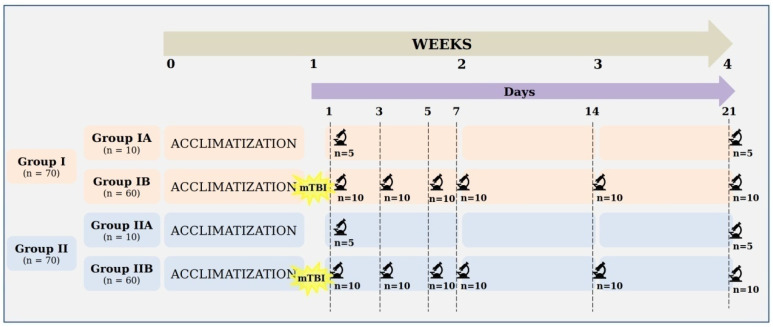
Detailed experimental design.

**Figure 2 cells-10-01121-f002:**
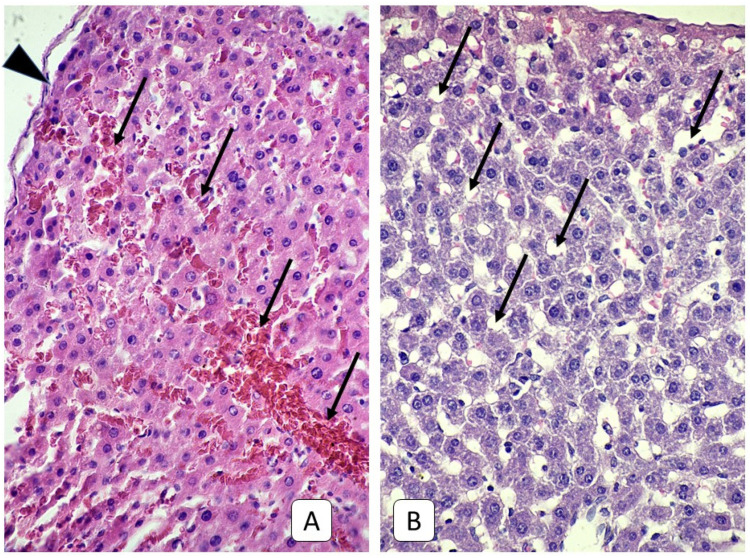
Photomicrograph of the liver. (**A**): adult rats, 1st day after mTBI. Dilated sinusoidal capillaries (arrow) near the liver capsule (arrowhead) with blood stasis and RBC sludges having occurred. (**B**): control group. Sinusoidal capillaries (arrow) of the subcapsular area. H E, ×400.

**Figure 3 cells-10-01121-f003:**
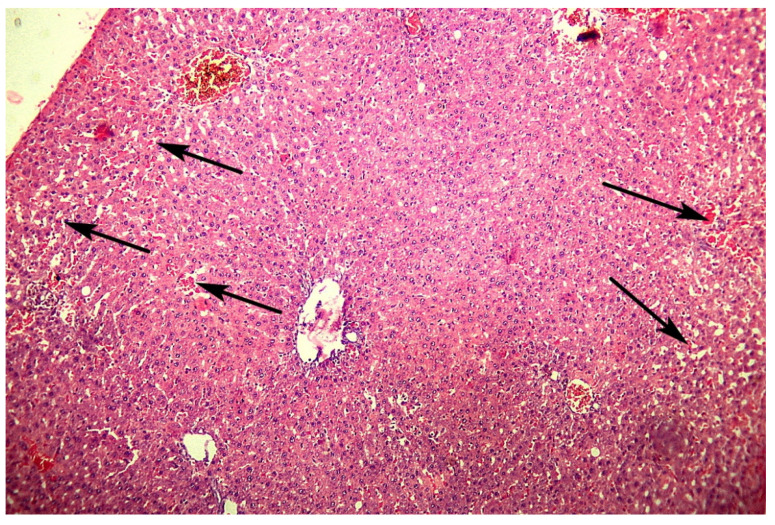
Photomicrograph of the liver. Adult rats, 3rd day after mTBI. Numerous dilated sinusoidal capillaries all over the visual field with blood stasis and RBC sludges having occurred. H E, ×100.

**Figure 4 cells-10-01121-f004:**
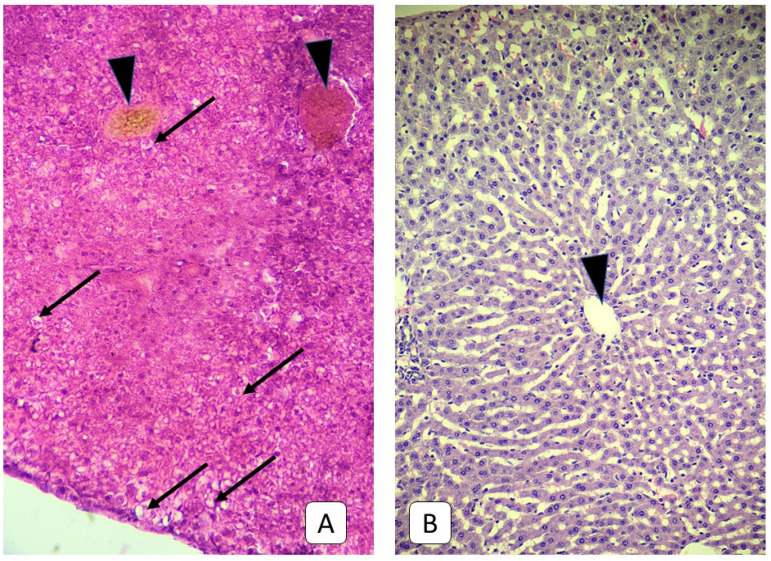
Photomicrograph of the liver. (**A**): juvenile rats, 3rd day after mTBI. A lot of dystrophic hepatocytes with swelling, light cytoplasm, and vacuoles near the liver capsule and central vein (arrow), the blood stasis in the central vein (arrowhead). (**B**): control group. Central vein (arrowhead) of the subcapsular area. H E, ×200.

**Figure 5 cells-10-01121-f005:**
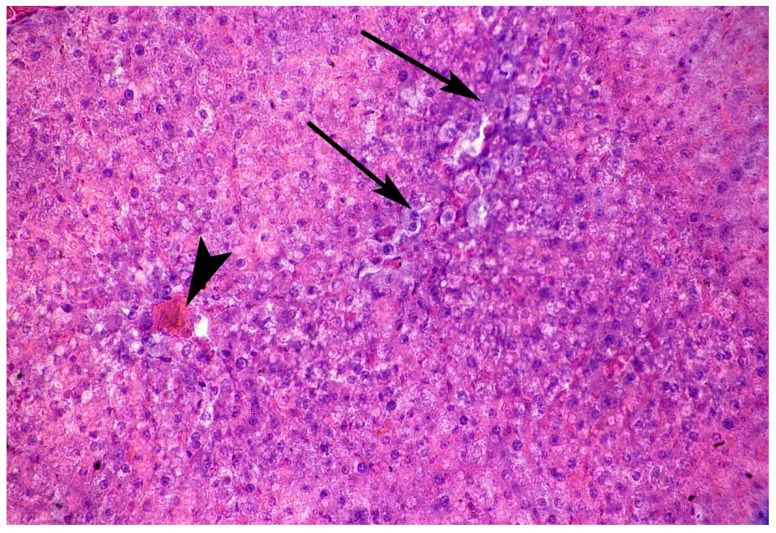
Photomicrograph of the liver. Juvenile rats, 5th day after mTBI. Focal necrosis and dystrophic hepatocytes with light cytoplasm and vacuoles (arrow), the blood stasis in the central vein (arrowhead). H E, ×400.

**Figure 6 cells-10-01121-f006:**
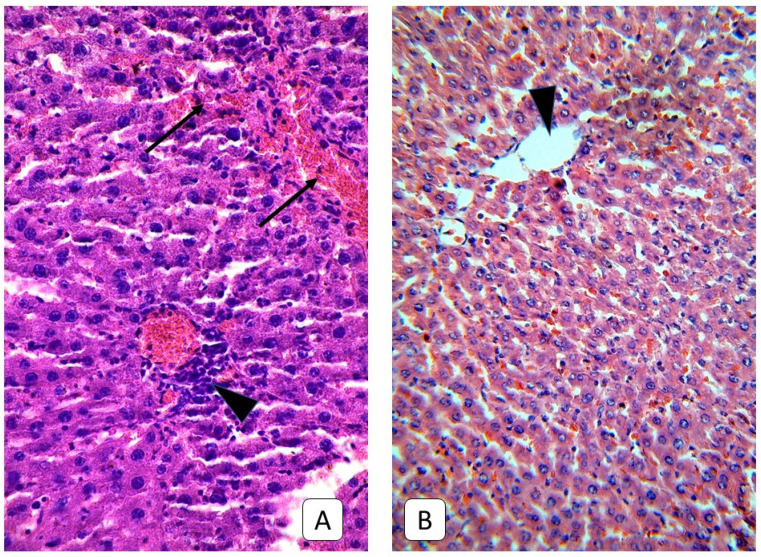
Photomicrograph of the liver. (**A**): adult rats, 7th day after mTBI. Dilated sinusoidal capillaries with RBC sludges and clots (arrow) and infiltrate near central vein (arrowhead). (**B**): control group. Central vein (arrowhead). H E, ×400.

**Figure 7 cells-10-01121-f007:**
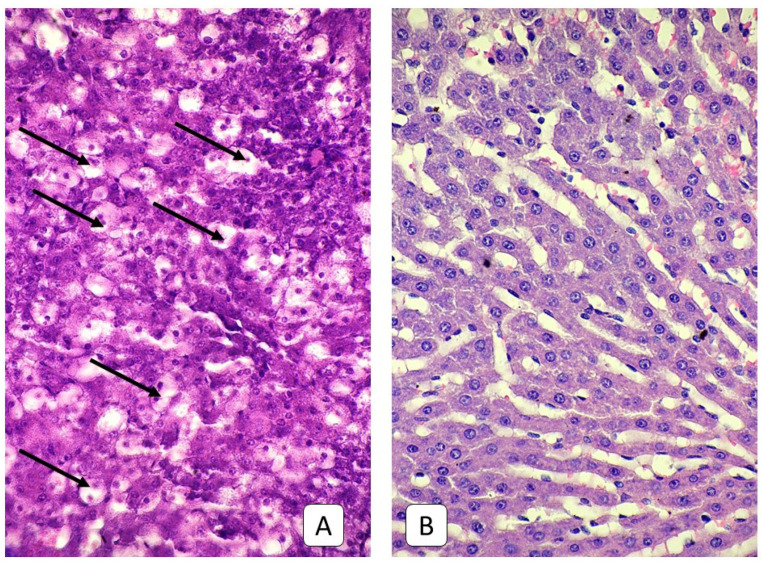
Photomicrograph of the liver. (**A**): juvenile rats, 7th day after mTBI. Dystrophic hepatocytes with swelling, light cytoplasm, pycnotic nuclei, and vacuoles (arrow). (**B**): control group. Liver parenchyma. H E, ×400.

**Figure 8 cells-10-01121-f008:**
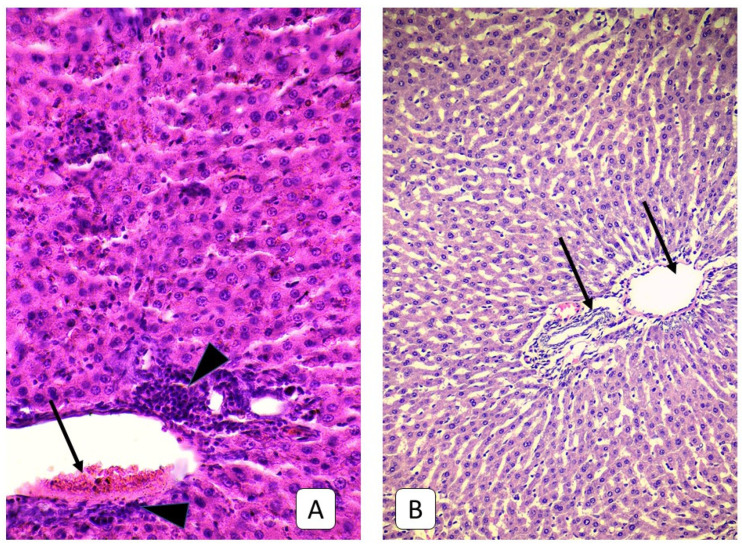
Photomicrograph of the liver. (**A**): adult rats, 21st day after mTBI. Periportal infiltrate (arrowhead) with collagen fibers and RBC adhesion to portal vein wall (arrow). (**B**): control group. Segmental portal triads (arrow). H E, ×200.

**Figure 9 cells-10-01121-f009:**
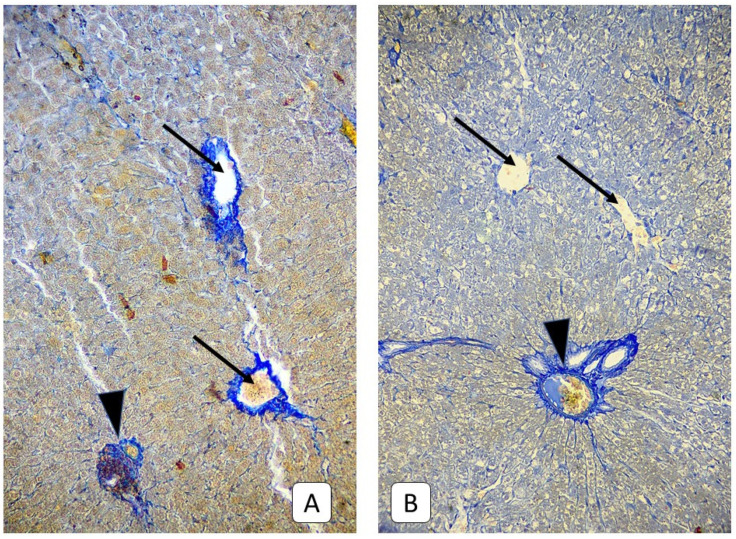
Photomicrograph of the liver. (**A**): adult rats, 21st day after mTBI. Collagen near central veins (arrow) and portal triads (arrowhead). (**B**): control group. Central veins (arrow) and collagen near portal triads (arrowhead) Mallory’s trichrome, ×400.

**Figure 10 cells-10-01121-f010:**
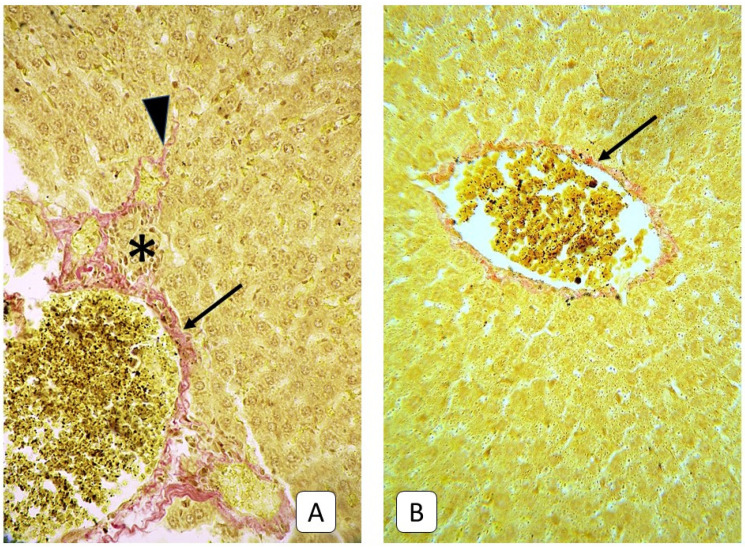
Photomicrograph of the liver. (**A**): adult rats, 21st day after mTBI. Periportal infiltrate (asterisk) with collagen fibers (arrow) and collagen invasion deeper to parenchyma (arrowhead). (**B**): control group. Collagen fibers in the wall of central vein (arrow). VanGiesone, ×400.

**Figure 11 cells-10-01121-f011:**
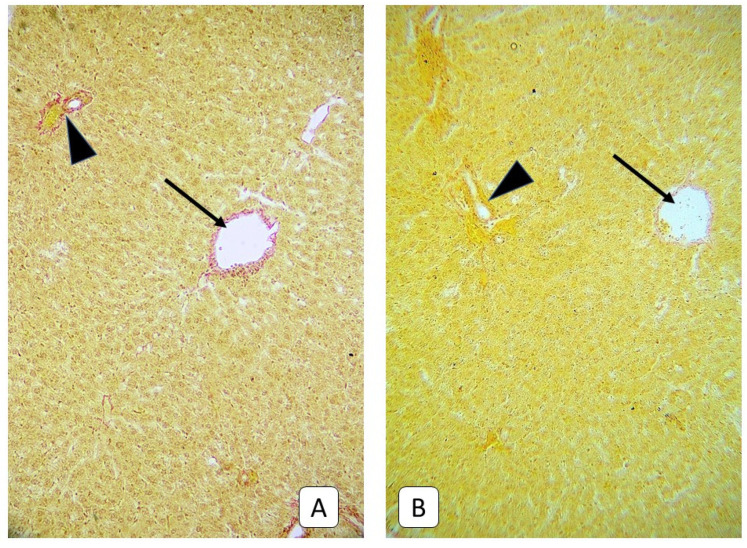
Photomicrograph of the liver. (**A**): juvenile rats, 21st day after mTBI. A thin layer of collagen fibers within the wall of central vein (arrow) and portal triad (arrowhead). (**B**): control group. Central vein (arrow) and portal triad (arrowhead). VanGiesone, ×400.

**Table 1 cells-10-01121-t001:** Summary of the morphometry of sinusoidal capillaries of the liver of adult rats and juvenile rats.

	Group IA [μm]	Numbers of Measurements	*p*	Group IIA [μm]	Numbers of Measurements	*p*
Control group	4.07, (3.56–4.68)	208		3.43, (3.06–4.09)	131	
Study group	**Group IB [μm]**			**Group IIB [μm]**		
Day 1	6.03, (4.44–8.69)	275	<0.001 *	3.72, (2.88–4.90)	188	0.61 *
Day 3	9.40, (7.62–11.82)	321	<0.001 * <0.001 ^#^	2.38, (1.92–2.85)	142	<0.001 * <0.001 ^#^
Day 5	10.17, (7.91–12.67)	246	<0.001 * 0.53 ^#^	2.64, (2.12–3.67)	126	<0.001 * 0.01 ^#^
Day 7	8.37, (6.16–11.01)	244	<0.001 * <0.001 ^#^	2.60, (2.14–3.44)	111	<0.001 * 0.99 ^#^
Day 14	9.53, (7.29–12.71)	277	<0.001 * 0.001 ^#^	3.16, (2.61–3.93)	111	0.052 * 0.052 ^#^
Day 21	7.78, (5.81–9.17)	205	<0.001 * <0.001 ^#^	3.19, (2.42–3.98)	111	0.052 * 0.99 ^#^

Note: Data presented as median, (Q1–Q3); *: the significance of difference compared with the control group; ^#^: the significance of the difference in comparison with the previous term of an experiment.

## Data Availability

Not applicable.

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
