# Peer review of "A Histological and Morphometric Assessment of the Adult and Juvenile Rat Livers after Mild Traumatic Brain Injury"

_cells, 2021, doi:10.3390/cells10051121_

Round 1

Reviewer 1 Report

This interesting article presents data obtained through an animal model to evaluate the histological and morphometric changes caused by mild traumatic brain injury (mTBI) in the liver tissue mTBI.

The paper is well written, even if I suggest a few minor changes before the publication.

The introduction section well introduced the thematic. I suggest clarifying the aims of the study in the last paragraph of this section.

The “Material and Methods” section should be improved.

In the first sub-section (2.1. Animals and Experimental Conditions), the authors should clarify that the experimentation has been performed following the animal care regulation. In the sub-section (2.3) figure 1: the timeline is expressed in weeks, while in the main text the authors inserted the time in days. Please, check it. Subsection (2.5): please, indicate clearly when you have applied t-test or ANOVA. It could be beneficial for the readers.

In the results section, table 1 should be improved, Inserting the statistical evaluation. Alternatively, you can insert a new table or in the main text or the supplementary file.

The discussion section should be improved, discussing the presented data with international data. Moreover, a brief paragraph should be inserted on the human application, in order to suggest future study.

Minor issue: English should be carefully checked. Please, check it.

Author Response

Respected Reviewer, Thank you very much for your review. The answers are in the attached file. 

Reviewer 2 Report

This is an interesting study of liver injury following traumatic brain injury (TBI). The authors provide only a histological evaluation; however, their findings are significant.

The findings would be improved if the authors would be able to provide additional histological information including data on hepatocyte and biliary/bile duct senescence and apoptosis. Since the authors use both young and aged models, the role of senescence would be important to understand in this context. 

Was there any hepatic fibrosis induced by TBI? again, this could be assessed by histological means, as well as both Kupffer cell activity (inflammation) and hepatic stellate cell activation - both hallmarks of liver damage following injury. 

Author Response

(The authors gave the same response as above.)

Reviewer 3 Report

The authors studied the experimental effects of mild traumatic brain injury (mTBI) on morphological changes in adult and juvenile rats' liver in a time-dependent model. The authors found drastic changes in the microcirculatory bed and inflammatory damage, while in juvenile rats, they observed dystrophy in the liver parenchyma. These changes appeared during the first 5-7 days of their study. Later, on 14-21 days, compensatory mechanisms were seen in both groups. The authors concluded that mTBI promotes deleterious modifications in the liver of both ages of rats and that they are different.

There are some suggestions and questions for the authors:

Format

  1. a) Central nervous system abbreviation should be CNS, not CSN. Correct 0.5 cm2 to cm2 in superscript, spaces between the words x400.bAll, and (Q1–Q3);*: the.

Background

  1. a) Data in Table 1 should also be presented as a bar or line graphic to observe better the two different time-course patterns between adult and juvenile rats. No units for morphometry measurements were given (mm?), although described in Materials and methods.
  2. b) In Figure 3, no arrows indicate the histological liver damage about sludge and blood stasis; otherwise, just trained pathologists could see them. Again, in Figure 4, pyknosis, micro or macro vacuoles, ballooning necrosis, etc., are not marked with arrows and numbers. Indeed, one or two panels of photographs should contain one representative picture for the samples of each day and comparing juvenile versus adult livers.
  3. c) The authors recognize the limitations of their work in the Discussion and Conclusion sections. Although this is a valuable novel approach that embraces histological and morphometry studies, the level of the journal with such impact factor requires the mandatory determination in plasma and liver of proinflammatory cytokines (TNF-alpha, IL-6, IL-1beta) by ELISA or immunohistochemistry just for hepatic samples, perhaps plasma and liver prostaglandins, and also oxidative stress markers as peroxidation by TBARS in hepatic tissues. These biochemical data would be correlated with changes observed by the authors, and the impact of their work would be enhanced to explain substantial differences between both ages in rats and the inflammatory mechanisms triggered.

Author Response

(The authors gave the same response as above.)

Round 2

Reviewer 2 Report

the authors have responded well to the critique

Author Response

Dear Reviewer,

Thank you very much for your opinion and all suggestions you gave us to make our manuscript better.

Reviewer 3 Report

The authors have answered the most requests and explained the budget limitations; they also, offered a deeper explanation in the Discussion section.

Author Response

(The authors gave the same response as above.)
